# Time diffraction-free transverse orbital angular momentum beams

Wei Chen [1,4 ✉], Wang Zhang[1,4], Yuan Liu[1], Fan-Chao Meng[2,3], John M. Dudley [2] & Yan-Qing Lu [1 ✉]

The discovery of optical transverse orbital angular momentum (OAM) has broadened our understanding of light and is expected to promote optics and other physics. However, some fundamental questions concerning the nature of such OAM remain, particularly whether they can survive from observed mode degradation and hold OAM values higher than 1. Here, we show that the strong degradation actually origins from inappropriate time-delayed $k_x$–$\omega$ modulation, instead, for transverse OAM having inherent space-time coupling, immediate modulation is necessary. Thus, using immediate $x$–$\omega$ modulation, we demonstrate theoretically and experimentally degradation-free spatiotemporal Bessel (STB) vortices with transverse OAM even beyond $10^2$. Remarkably, we observe a time-symmetrical evolution, verifying pure time diffraction on transverse OAM beams. More importantly, we quantify such nontrivial evolution as an intrinsic dispersion factor, opening the door towards time diffraction-free STB vortices via dispersion engineering. Our results may find analogues in other physical systems, such as surface plasmon-polaritons, superfluids, and Bose-Einstein condensates.

[1] National Laboratory of Solid State Microstructures, Key Laboratory of Intelligent Optical Sensing and Manipulation, College of Engineering and Applied Sciences, and Collaborative Innovation Center of Advanced Microstructures, Nanjing University, Nanjing 210093, China. [2] Institut FEMTO-ST, Université Bourgogne Franche-Comté CNRS UMR 6174, Besançon 25000, France. [3] State Key Laboratory of Integrated Optoelectronics, College of Electronic Science and Engineering, Jilin University, 2699 Qianjin Street, Changchun 130012, China. [4]These authors contributed equally: Wei Chen, Wang Zhang. ✉email: wchen@nju.edu.cn; yqlu@nju.edu.cn

Vortex structures that carry orbital angular momentum (OAM) have been the subject of extensive research and applied to diverse areas, including optics[1–4], acoustics[5–7], and electronics[8–10], both in classical and quantum regions. The best-known optical vortices with OAM can be generated by introducing a spiral phase into the two-dimensional transverse plane (i.e., the $x–y$ plane) of light fields. Note that such OAM is generally longitudinal, which means the OAM vector is parallel to the propagation direction $z$. Using a similar manner, one can conveniently generate longitudinal OAM beams in other physical systems.

Recently, there is rapidly growing interest[11–20] in transverse OAM where the OAM vector is orthogonal to the propagation direction. Such transverse vortices are not uncommon in nature and science. As early as 1770, Captain Cook discovered a similar phenomenon in the intriguing movement of the boomerang used by the First Nations Australian peoples. The tropical cyclone, a transversely moving vortex flow, has always been a research hotspot in meteorology and even economics[21]. Similar vortex flow has also been observed in human left ventricular blood flow, which can be used to diagnose and assess ventricular contractility[22]. In magnetic nanowires, controlling the formation and movement of vortex domain walls at the nanoscale is a fertile ground to explore emergent phenomena and their technological prospects, such as microelectronic devices and robust memory systems[23,24]. In optics, such transverse OAM was first observed in femtosecond filaments in air[12] and was subsequently realized in polychromatic wave packets in free space[13,14], known as spatiotemporal (ST) optical vortices. This transverse OAM beam shows great potential to extend the applications of longitudinal OAM and to promote optics or other basic physics. However, previous studies still have some limitations. For instance, although a few studies suggested a strong ST coupling in ST vortices compared with their longitudinal counterparts, a precise description and accurate analysis of this nontrivial coupling mechanism are still lacking. Most importantly, existing experiments were limited in low transverse OAM, e.g., $l = 1$, 2 (where $l$ is the topological charges), in which a 2-order ST vortex rapidly degrades into two first-order ST vortices during propagation[13,14] or nonlinear interaction[15,16]. Particularly, such degradation leads to instability of integral OAM value[16], limiting greatly the scientific research for and the real-world application of transverse OAM. Till now, most experimental efforts attributed this mode degradation to ST astigmatism effect, i.e., the mismatch between spatial diffraction and medium dispersion[14,16,19]. In contrast, a few theoretical studies introduced a novel effect—termed time diffraction—via a plane-wave expansion method, providing a new perspective to investigate ST vortices and their degradation[11,17]. Notably, ST astigmatism and temporal diffraction appear to be distinct because the former depends on the medium, while the latter is intrinsic, similar to spatial diffraction of monochromatic beams.

Here, based on wavevector analysis, the nontrivial features and inherent ST coupling of transverse OAM are uncovered. We show that the time-delayed $k_x–\omega$ modulation relies on a spatial Fourier transform (SFT) in current experiments actually made the most contributions to the observed mode degradation, which could be circumvented by an immediate $x–\omega$ modulation via the inverse design of phase. This allows us to generate theoretically equivalent spatiotemporal Bessel (STB) vortices and observe ultrahigh transverse OAM even beyond $10^2$. We also show that STB vortices behave an opposite time-symmetrical evolution with respect to signs of $l$ due to therein inherent ST coupling. By circumventing the modulation-induced degradation, we confirm that such a never-seen-before phenomenon can be explained perfectly by time diffraction, with no need for considering the ST astigmatism. Beyond, we further show that such nontrivial ST coupling can be quantified as an intrinsic dispersion factor, and thus, theoretically, can be compensated by the medium dispersion thereby obtaining time diffraction-free STB vortices. Our work paves the way for further research and application of this unique OAM.

## Results

**Nontrivial features and inherent ST coupling of transverse OAM.** To better understand why the transverse OAM is less trivial than the longitudinal case, let us re-exam the difference between these two situations. Generally, the angular momentum density of optical fields can be written as[1] $\mathbf{L} = \mathbf{r} \times \mathbf{p}$, where $\mathbf{r}$ is position vector and $\mathbf{p}$ is linear momentum density associated with a local wavevector $\mathbf{k}_l$. For a vortex beam, the carried OAM should remain stable during propagation. This means that the field with the same position vector $\mathbf{r}_0$ should move as a whole, and thus they share the same propagation vector $\mathbf{k}_z$ and the linear momentum density with the same amplitude $|\mathbf{p}|$, i.e., a local wavevector $|\mathbf{k}_l|$.

For the longitudinal OAM beam, the local wavevector $\mathbf{k}_l$ lies on the $x–y$ plane and is always perpendicular to $\mathbf{k}_z$ during propagation (Fig. 1a), i.e., $\mathbf{k}_l \perp \mathbf{k}_z$. One sees immediately that, a monochromatic light field can naturally meet the demand, that is, $k_0 = \sqrt{|\mathbf{k}_l|^2 + |\mathbf{k}_z|^2} \equiv \text{const}$, where $k_0$ is the whole wavevector. While for the transverse OAM beam, the $\mathbf{k}_l$ plane is now rotated by 90 degrees to follow the rotation of the OAM vector, becoming the $x–z$ plane (Fig. 1b). Note that $\mathbf{k}_l$ is no longer perpendicular to $\mathbf{k}_z$. If the monochromaticity is still forcibly retained, another wavevector $\mathbf{k}_y\ (\mathbf{r}_0)$ varying along the ring is required to compensate for the wavevector mismatch, that is, $|\mathbf{k}_y(\mathbf{r}_0)| = |\Delta\mathbf{k}(\mathbf{r}_0)| = \sqrt{k_0^2 - |\mathbf{k}(\mathbf{r}_0)|^2}$. Obviously, such conditions are difficult to satisfy, at least experimentally. Nevertheless, if we turn to the polychromatic field, $\Delta\mathbf{k}(\mathbf{r}_0)$ can be easily compensated by a broadened time spectrum $\underline{\Delta\omega}$, i.e., $k(\omega) = k(\mathbf{r}_0) = \sqrt{|\mathbf{k}_l(\mathbf{r}_0)|^2 + |\mathbf{k}_z|^2 + 2|\mathbf{k}_l(\mathbf{r}_0)||\mathbf{k}_z|\cos[\vartheta(t)]}$. This explains why transverse OAM was first found in femtosecond filaments, and polychromatic wave packets become the possible solution for carrying transverse OAM. Most importantly, the strong interaction between $\mathbf{k}_l$ and $\mathbf{k}_z$ leads to an inherent ST coupling in transverse OAM beams, indicating their nontrivial time-varying features. This coupling is also reflected in the specific ST frequency-frequency relationship inside an ST vortex, as we will demonstrate below[25–27].

**From time-delayed $k_x–\omega$ modulation to immediate $x–\omega$ modulation.** The above discussion also implies that the way to generate an ST vortex must be immediate. In contrast, any time-delayed modulation inevitably resonates with the inherent ST coupling within an ST vortex, resulting in strong mode degradation. We note that all recent experiments have been realized by directly loading a spiral phase via a conventional pulse shaper[28] that contains a phase device between two gratings in a $4f$ system. In this arrangement, the phase-loaded plane is regarded as $k_x–\omega$ plane and the ST vortex generally corresponds to a "patch" on the light-cone (Fig. 1c). Naturally, such a scheme is inspired by the generation of longitudinal OAM vortices. Because the $4f$-shaper is only used to realize temporal modulation, an additional SFT is required to convert the spiral phase onto $x–t$ plane, which was realized by a cylindrical lens or free-space transmission. Its principle is, essentially, to use the transmission to achieve the SFT; obviously, such approach is time-delayed which works well around a certain position—typically, the Fourier plane of the cylindrical lens or the far field—and the ST vortices inevitably

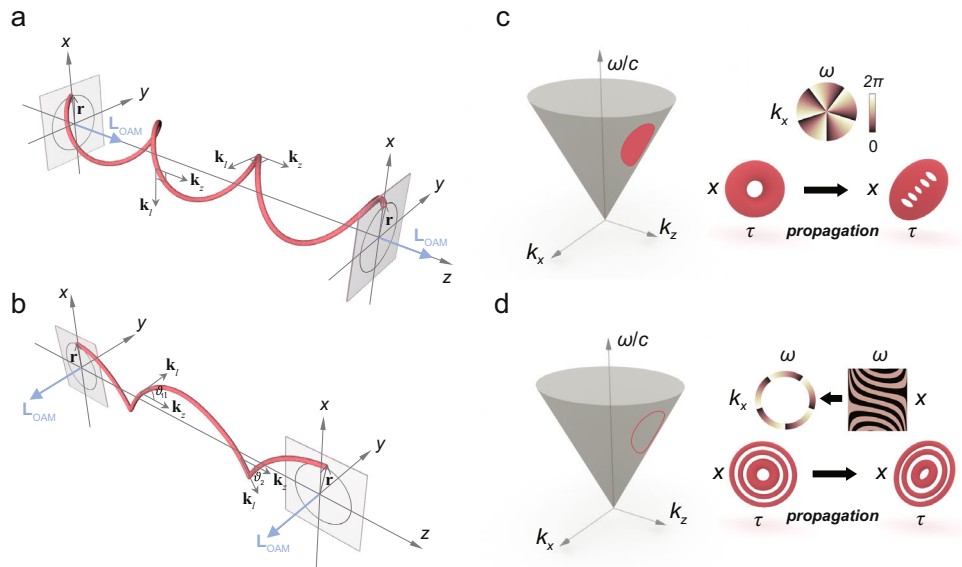

**Fig. 1 Wavevector analysis of longitudinal and transverse OAM and two schemes for transverse OAM beams generation. a** For a monochromatic longitudinal OAM beam, the local wavevector $\mathbf{k}_l$ is always perpendicular to the propagation vector $\mathbf{k}_z$. **b** For a transverse OAM beam, the monochromaticity is broken due to the interaction between the local wavevector $\mathbf{k}_l$ and the propagation vector $\mathbf{k}_z$, resulting in the generation of polychromatic ST vortex with inherent ST coupling. Note that spiral and cycloid curve in **a**, **b** represent the linear momentum vectors **p** respectively, which donate the OAM by $\mathbf{L} = \mathbf{r} \times \mathbf{p}$. **c** An Gaussian-like ST vortex with $l = 5$ corresponds to a "patch" with a spiral phase on the light cone $k_x^2 + k_z^2 = (\omega/c)^2$. Notably, the current scheme cannot effectively produce such a spiral phase on the $k_x$–$\omega$ plane due to the time-delayed modulation, and the 5-order ST vortex degrades into five first-order ST vortices rapidly. **d** Inverse design of the phase makes it possible to immediately produce an impulse ring with a spiral phase on the $k_x$–$\omega$ plane, leading to the generation of a degradation-free 5-order STB vortex.

degrade at other positions (Fig. 1c). This explains the observed strong degradation in recent experiments, especially for high-order vortices.

In this work, we show that according to the inverse design of the spiral phase, the time-delayed SFT is no longer required, thereby realizing a degradation-free ST vortex directly from immediate $x$–$\omega$ modulation, where "immediate" means that the SFT is pre-processed on the phase pattern, with no need for extra propagation. Therefore, one can use the conventional liquid-crystal (LC) based spatial light modulators (SLM) to accomplish such modulation, although their response is not fast. Such operation corresponds to an impulse ring with a spiral phase on the $k_x$–$\omega$ plane (as shown in Fig. 1d), i.e.,

$$\tilde{E}(k_x, \Omega) = \tilde{E}_l(\kappa, \varphi) = \delta(\kappa - R_0)e^{il\varphi}, \tag{1}$$

where $\Omega = \gamma(\omega - \omega_0)$ relates to the detuning frequency, $\omega_0$ is the central frequency, $\gamma = \Delta k_x / \Delta\omega$ is the reduction coefficient for temporal and spatial scale consistency, $\kappa = \sqrt{k_x^2 + \Omega^2}$ and $\varphi = \arctan(k_x/\Omega)$ are the polar coordinates on the $k_x$–$\omega$ plane, and $R_0$ is the modulated radius. Equation (1) also describes the strong correlation between spatial frequency $k_x$ and temporal frequency $\omega$ inside this beam, implying that changing the spectral properties also changes its spatial properties. The field on the $x$–$t$ plane can be calculated by a two-dimensional Fourier transform (Supplementary Note 1):

$$
\begin{aligned}
E_l(\rho, \theta) &= \mathrm{F \cdot T \cdot}\left[\tilde{E}_l(\kappa, \varphi)\right] \\
&= \frac{1}{2\pi}\int_0^{2\pi}\int_0^\infty \delta(\kappa - R_0)e^{il\varphi}e^{i\kappa\rho\cos(\varphi-\theta)}\kappa\,d\kappa\,d\varphi \\
&= (i)^l R_0 e^{il\theta}J_l(R_0\rho),
\end{aligned}
\tag{2}
$$

where $\rho = \sqrt{x^2 + \tau^2}$ and $\theta = \arctan(x/\tau)$ are the polar coordinates on the $x$–$t$ plane, $J_l$ is the $l$-order Bessel function of the first kind, and $\tau = t - z/v_g$ is the retarded time in pulse frame where $v_g$

is the group velocity. Equation (2) shows that the generated fields are strictly equivalent to the optical STB vortices, the exact solutions of the paraxial wave equation described by Dallaire et al.[29] in 2009. Actually, it was not until 2012 that Bliokh and Nori[11] theoretically pointed out that these beams carry transverse OAM. Their very recent work[17] theoretically proposed an observable spin-orbit interaction between the transverse OAM and spin[30–32] in such beams, providing an intersection of these two hot topics. However, due to the similar issue of modulation method, these beams lack experimental observation so far. We note that a strategy to synthesize STB vortices via superimposing a spiral phase with a conical phase has been proposed very recently[19]. Still, due to its reliance on time-delayed modulation, obvious separation of topological charges even for $l = 2$ has been observed.

To achieve such STB vortices via $x$–$\omega$ phase modulation, the spiral phases are inversely designed, inspired by the fact that the space coordinate $x$ and the spatial frequency $k_x$ are essentially a Fourier transform pair. Therefore, one can project a single point $k_n e^{i\phi_n}$ in the spatial frequency domain onto a location-shifted grating $G(x) = (x - x_n)/\Lambda_n \mod 2\pi$ in the $x$ axis, wherein the period and shifted displacement of the grating are $\Lambda_n = 2\pi/k_n$ and $\Delta x_n = \phi_n/k_n$, respectively (Supplementary Fig. 1). The phases for generating STB vortices with topological charges of $l = 10, 25, 50,$ and $100$ are shown in Fig. 2a–d, in which topological charges are manifested as the amount of dislocation between the left and right main lobes in the phase diagram.

**STB vortices with ultrahigh transverse OAM even beyond $10^2$.** To generate STB vortices, we used a custom $4f$-pulse shaper consisting of a diffraction grating (1800 lines/mm, GH25-18V, Thorlabs), a cylindrical lens ($L_{y1}$ with a focal length of $f = 100$ mm, which also determines the distances between the elements), and an LC-based 2D phase-only SLM (PLUTO-2.1-

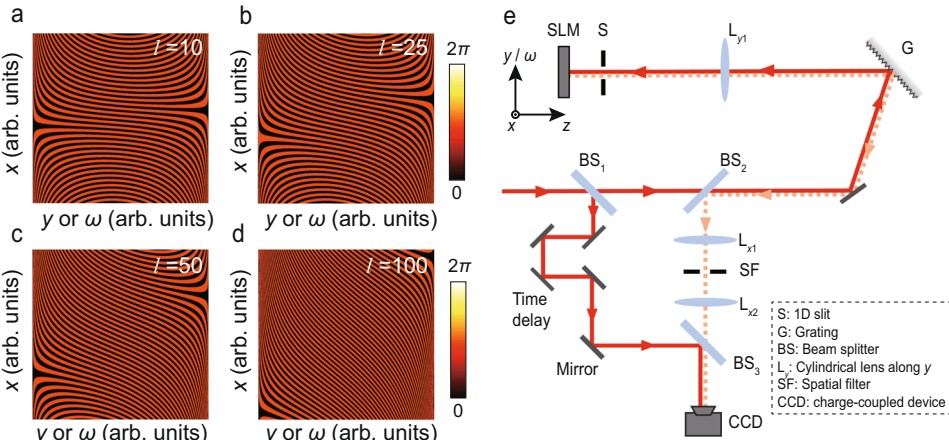

**Fig. 2 SLM phase patterns and experimental setup for generating and characterizing STB vortices. a–d** Phase patterns of STB vortices with topological charges of $l = 10$, 25, 50, and 100. **e** Experimental setup consists of two sections: (1) STB vortex generator consisting of a grating, a cylindrical lens $L_{y1}$, an aperture, and an SLM; and (2) a time-resolved profile analyser that is realized by a Mach–Zehnder interferometer consisting of two BSes (BS$_1$ and BS$_3$), a CCD camera, and a motorized translation stage in the reference path. The coordinate in the upper left corner of **e** marks the orientations in SLM where the phase patterns in **a–d** are loaded.

NIR-133, Holoeye) with a resolution of $1920 \times 1080$, a pixel pitch of ~8 μm, and an active area of ~$15.36 \times 8.64$ mm (Fig. 2e). The frequencies of an ultrashort pulse that comes from a Ti:sapphire laser with a central wavelength of ~800 nm and a pulse duration of ~35 fs are spatially spread by the grating and collimated to the SLM via the $L_{y1}$, which can be understood as a temporal Fourier transform. We consider the SLM as the $x$–$\omega$ plane, where the phase patterns for the generation of STB vortices are loaded. After the SLM, the light field is retroreflected and reconstituted at the same diffraction grating, thereby immediately generating the STB vortices, with no need for a time-delayed SFT. The ST intensities were measured and reconstructed using a Mach–Zehnder interferometer as shown in Fig. 2e (Methods)[33,34].

The experimental and simulated (Methods) results of STB vortices with $l = 10$, 25, 50, and 100 are shown in Fig. 3. The reconstructed intensities are plotted in Fig. 3a–d, and simulations are drawn with respective illustrations accordingly. Notably, the spatial and temporal bandwidths were set to $\Delta k_x = $ ~123 rad/mm (by the phase patterns) and $\Delta \lambda = $ ~12 nm (by the one-dimensional slit with a width of ~5 mm (Fig. 2e)) for temporal and spatial scale consistency, i.e., the standard STB vortices should be circularly symmetric pulses with equal width and length (when we project $t$ axis to $z$ axis) to make the integral OAM value is $l\hbar$ per photon, otherwise, the OAM value will become larger[17]. The corresponding $\gamma$ is ~3.48 ns/m. Despite the slight distortion, the experimental results are in good agreement with the simulations. To ensure that SLM does not affect the polarization, we adjusted the incident light to be linearly polarized in the $y$ direction, which means that the polarization of the generated STB vortices is perpendicular to the $x$–$t$ plane. According to the theoretical analysis in ref. [17], this out-of-plane polarization will result in a zero intensity at the centre of STB vortices (verified by Fig. 3a–d). In contrast, the in-plane polarization will lead to an observable nonzero intensity in the centre of STB vortices due to transverse spin-orbit interaction[17].

Besides, the phase reconstruction of high-order modes is difficult due to the instability of the interferometer caused by vibrations such as air disturbance (Methods). Nevertheless, the simulated phases (Fig. 3e–h) verify the spiral phases of the corresponding topological charges, implying the carried transverse OAM (see also the reconstructed phases with $l = 3$, 5, 10, and 25 shown in Supplementary Fig. 3). The reconstructed three-

dimensional intensity iso-surface profiles of these beams from measured data are shown in Fig. 3i–l (Methods). Furthermore, we calculated the spatial and temporal diameters of these beams from Eq. (2). The theoretical, simulated, and experimental results as a function of topological charges are shown in Fig. 3m, n, indicating that their diameters are linearly related to the topological charge, except for $l = 0$. Additionally, the widths of the STB vortices on the $x$ and $t$ axes are inversely proportional to the spatial and temporal spectral bandwidths (see also Supplementary Fig. 3), respectively, which are consistent with conventional spatial beams and short pulses. Enabled by the proposed $x$–$\omega$ modulation, our results provide the experimental observation of ultrahigh transverse OAM even beyond $10^2$, which is two orders of magnitude higher than the results reported thus far.

**Quantifying ST coupling by an intrinsic dispersion factor $\beta_2^{int}$.** To investigate the propagation dynamics, we captured two STB vortices with opposite topological charges of $l = 100$ and $-100$ at different positions along $z$. The intensities with $l = 100$ at $z = -150$, $-100$, $-50$, 0, 50, 100, and 150 mm are shown in Fig. 4a. We marked the position of a standard mode (described in Eq. (2)) as $z = 0$ mm and observed a time-symmetrical evolution on the $t$ axis. Remarkably, unlike the longitudinal OAM beam, for which the sign of the topological charge is difficult to judge without extra devices, such as the cylindrical lens, the results with $l = -100$ are distinguishable because they look like the mirror version of $l = 100$ (Fig. 4b). Since the modulation-induced degradation is circumvented, we immediately realized that such unique evolution could be explained by the time diffraction, which describes the accumulated phase difference in STB vortices between waves with different frequencies[11]. In this model, STB vortices could be considered as the coherent superposition of a series of plane waves[11],

$$\psi(\mathbf{r}, t) \propto \int_0^{2\pi} e^{i[k_0 z + \Delta k \cos \tilde{\phi} z + \Delta k \sin \tilde{\phi} x + l\tilde{\phi} - \omega(\tilde{\phi})t]} d\tilde{\phi}, \qquad (3)$$

where $\omega(\tilde{\phi}) = c\sqrt{k_0^2 + \Delta k^2 + 2k_0 \Delta k \cos \tilde{\phi}}$, $c$ is the speed of light in a vacuum, $\Delta k$ is the spatial bandwidths, and $\psi(\mathbf{r}, t)$ is the electric field. As shown in Fig. 4c, the results by calculating intensities $|\psi(\mathbf{r}, t)|^2$ with the same parameters in our experiment verified our observation of pure time diffraction.

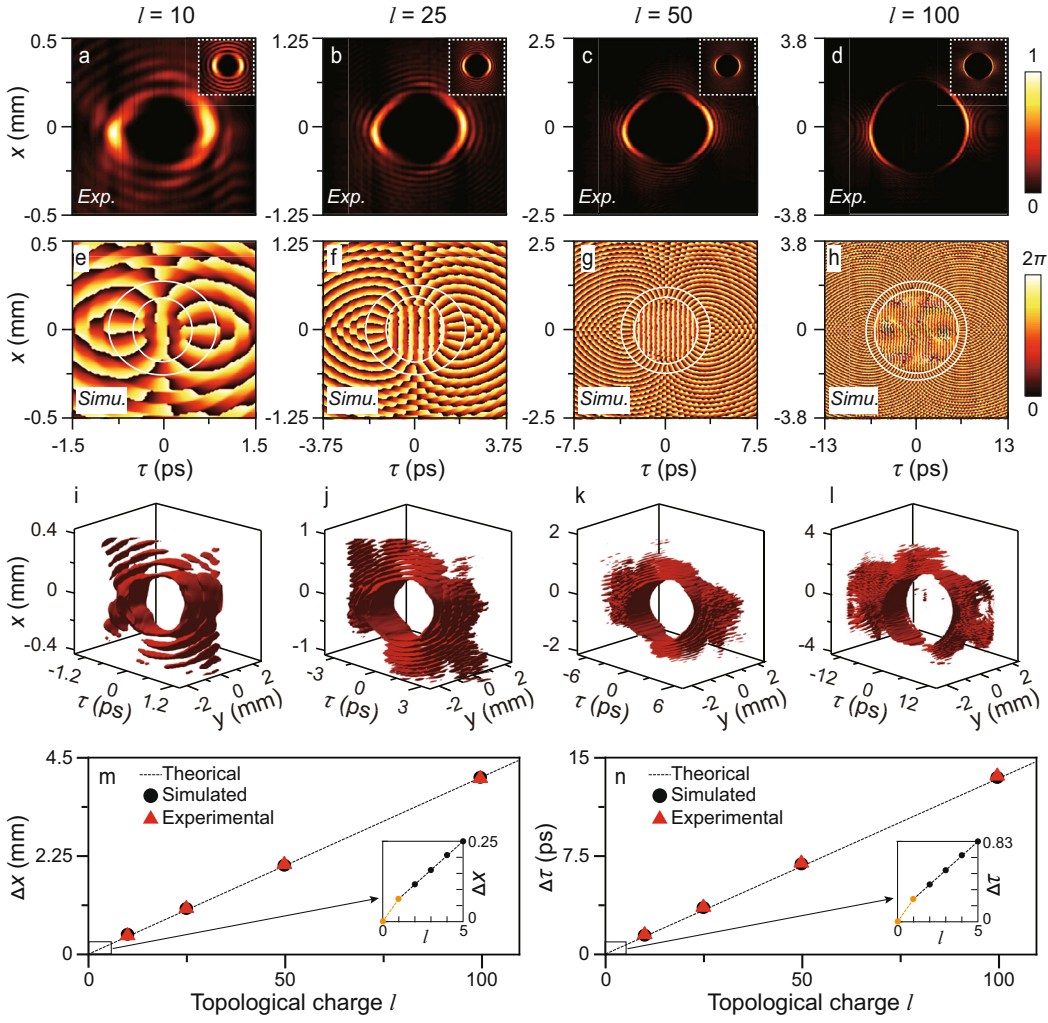

**Fig. 3 Experimental and simulated results of STB vortices with topological charges of $l = 10, 25, 50,$ and 100. a–d** Reconstructed intensities.
**e–h** Simulated phases. **i–l** Reconstructed 3D profiles. **m, n** Spatial or temporal diameter dependence of STB vortices. The white dotted squares (circles) in
**a–d** (**e–h**) show the simulated intensities (spiral phases) of each STB vortex. The corresponding topological charges are marked at the top of each column.
Exp: experimental results; Simu: simulated results.

We note that the time diffraction actually acts on STB vortices in a similar manner as the medium dispersion acts on optical pulses[35]. Indeed, by setting a pre-chirp to ~−0.231 ps², we observed consistent results in the simulation (Fig. 4d), which means the pre-chirp is completely compensated after a short distance of propagation from $z = -150$ mm (the elliptical chirped STB vortex) to $z = 0$ mm (the standard non-chirped STB vortex). At first, we attributed this to the positive dispersion of the air, however, the corresponding dispersion factor $\beta_2$ is ~1.542 ps²/m—~60,000 times the second-order dispersion of air—which implies that this dispersion should be intrinsic. Inspired by the inherent wavevector interaction within transverse OAM beams, we re-examine the ST spectra of STB vortices as described in Eq. (1), where the spatial frequencies $k_x$ and temporal frequencies $\omega$ actually hold a one-to-one correspondence. By combining Eq. (1) with the light cone equation $k_x^2 + k_z^2 = (\omega/c)^2$, we can obtain the relation between the propagation constant $k_z$ and $\omega$. Formally, an intrinsic dispersion factor $\beta_2^{int}$ could be obtained by Taylor series expansion of $k_z(\omega)$, i.e., $\beta_2^{int} = \partial^2 k_z(\omega)/\partial\omega^2\big|_{\omega=\omega_0}$[35]. In this manner, the dispersion factor could be written as

(Supplementary Note 5)

$$\beta_2^{int}(\omega_0) = \left[\gamma^2\left(\frac{\omega_0^2}{c^2} - R_0^2\right) - \frac{R_0^2}{c^2}\right]\Big/\left(\frac{\omega_0^2}{c^2} - R_0^2\right)^{\frac{3}{2}}. \quad (4)$$

Using Eq. (4), we directly obtain $\beta_2^{int} =$ ~1.542 ps²/m, which is perfectly consistent with the dispersion factor estimated by our simulation. This makes it clear that the inherent ST coupling in STB vortices can be quantified as an intrinsic dispersion factor $\beta_2^{int}$. Base on this, we propose a group dispersion delay (GDD) model to provide a theory-accurately and engineering-friendly description of STB vortices. For simplicity, we only consider the second-order dispersion and ignore the higher-order terms. In this model, the ST spectra of STB vortices could be rewritten as $\tilde{E}_l(\kappa, \varphi, z) = \delta(\kappa - R_0)e^{il\varphi}e^{\frac{i}{2}\beta_2^{int}z(\kappa\cos\varphi)^2}$, where the factor $e^{\frac{i}{2}\beta_2^{int}z(\kappa\cos\varphi)^2}$ donates the GDD. Therefore, the fields on the $x$–$t$ plane become

$$E_l(\rho, \theta; z) = \frac{1}{2\pi}\int_0^{2\pi}\int_0^{\infty}\delta(\kappa - R_0)e^{il\varphi}e^{\frac{i}{2}\beta_2^{int}z(\kappa\cos\varphi)^2}e^{i\kappa\rho\cos(\varphi-\theta)}\kappa d\kappa d\varphi. \quad (5)$$

As expected, by solving Eq. (5) numerically with $\beta_2^{int} =$ 1.542 ps²/m, the results are perfectly consistent with the

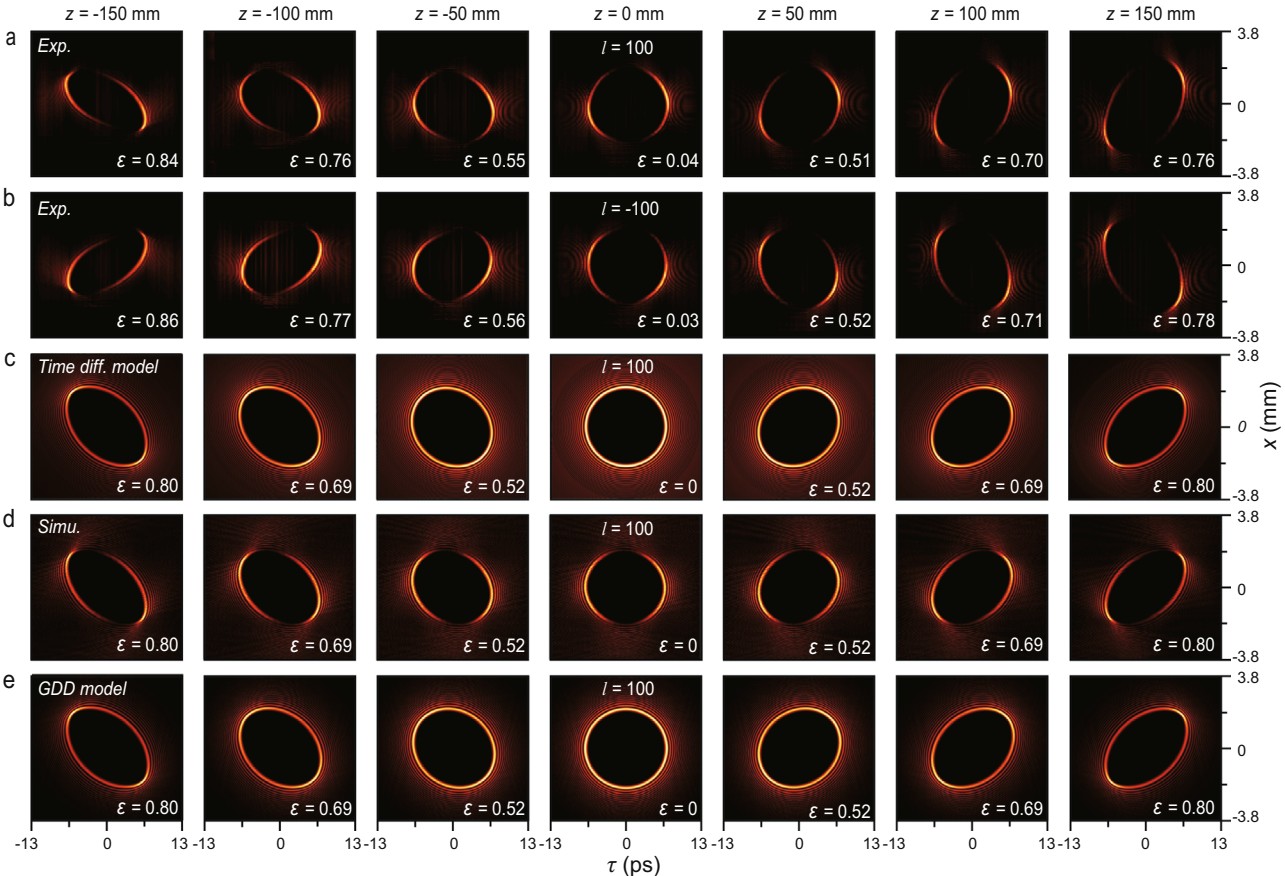

**Fig. 4 Experimental, theorical, and simulated results of propagation dynamics of two STB vortices with topological charges of $l = 100$ and $-100$.**
**a** Reconstructed intensities of an STB vortex with the topological charge of $l = 100$ at $z = -150, -100, -50, 0, 50, 100$, and 150 mm, where the position of the standard mode (described in Eq. (2)) is marked as $z = 0$ mm. **b** Same as **a**, but with a topological charge of $l = -100$. **c** Theorical calculated results of **a** using the time diffraction model. **d** Simulated results of **a** with a pre-chirp of $\sim -0.231$ ps². **e** Theorical calculated results of **a** using the GDD model with an intrinsic dispersion factor $\beta_2^{int} = \sim 1.542$ ps²/m. The corresponding eccentricities are given in the lower right corner of each figure in **a–e**. The positions are marked at the top of each column. Time diff. model: time diffraction model.

experiment, time diffraction model, and the simulation, as shown in Fig. 4e.

Additionally, we calculated the eccentricity of these two beams at different positions to quantify and compare the mode distortions caused by the time diffraction, i.e., $\varepsilon = \sqrt{1 - (\sigma/\eta)^2}$, where $\sigma$ and $\eta$ are the major and minor axes of these beams on the $x$–$z$ plane. The farther the value of $\varepsilon$ is from 0—corresponding to a standard circle—the larger the distortion of the mode is. The evolution of eccentricity further verifies the time-symmetrical evolution, despite the minor deviation in the experiment owing to the slight beam divergence caused by incomplete collimation.

**Towards time diffraction-free STB vortices.** We next try to eliminate the time diffraction as much as possible. One way that has been theoretically proposed is to produce the so-called Lorentz-boosted STB vortices[11,17], however, such beams are difficult to generate experimentally because of the need to give a monochromatic Bessel vortex an extra speed along the $y$ direction. Another rough method is just to simultaneously reduce the time and space bandwidth, i.e., $\Delta\lambda$ and $\Delta k_x$, akin to the fact that pulses with narrower bandwidths are less sensitive to the dispersion. Here, based on the GDD model, we propose that by engineering the dispersion of media, the time diffraction can be completely suppressed, thereby achieving time diffraction-free STB vortices.

The simulated evolutions of an STB vortex with $l = 100$ propagating in virtual media with the second-order dispersion of $-0.3\beta_2^{int}$ and $-0.9\beta_2^{int}$ are shown in Fig. 5a, b, respectively, where the time diffraction is suppressed to varying degrees. We summarize the calculated integral OAM values (Methods) and eccentricities at different propagating distances with media dispersions varying from $-0.3\beta_2^{int}$ to $-0.99\beta_2^{int}$ (Fig. 5c). Naturally, one can obtain STB vortices propagating in a stable and invariant manner if the high-order dispersion terms are further compensated. Additionally, we find that integral OAM values in Fig. 5c can be approximately expressed as $\langle L(z)\rangle \approx \{1 - [\omega_0^2\alpha(l)/4\pi^2](\beta_2^{int} + \beta_2^{ext})|z|\}l$, where $l$ is the OAM value of the standard STB vortex, $\alpha(l)$ is a dimensionless constant related to $l$ where $\alpha(l = 100) \approx \sqrt{2}\times 10^{-6}$, $\beta_2^{ext}$ is the media dispersion, and $|z| = |ct|$ is the propagation distance from the standard location. The time-varying transverse OAM $\langle L(z)\rangle$ may be considered as a special physical quantity to define ST vortices because it indicates that their OAM fundamentally depends on the intrinsic dispersion factor $\beta_2^{int}$. It would be interesting and meaningful to derive a more precise analytical expression of $\langle L(z)\rangle$ and find other unique physical quantities of the ST vortices in the future.

Furthermore, it turns out that the higher-order STB vortices show less sensitivity to time diffraction, by investigating the evolutions of STB vortices with different topological charges, i.e., $l = 25$ and 0. For $l = 25$, the time-symmetrical evolution was

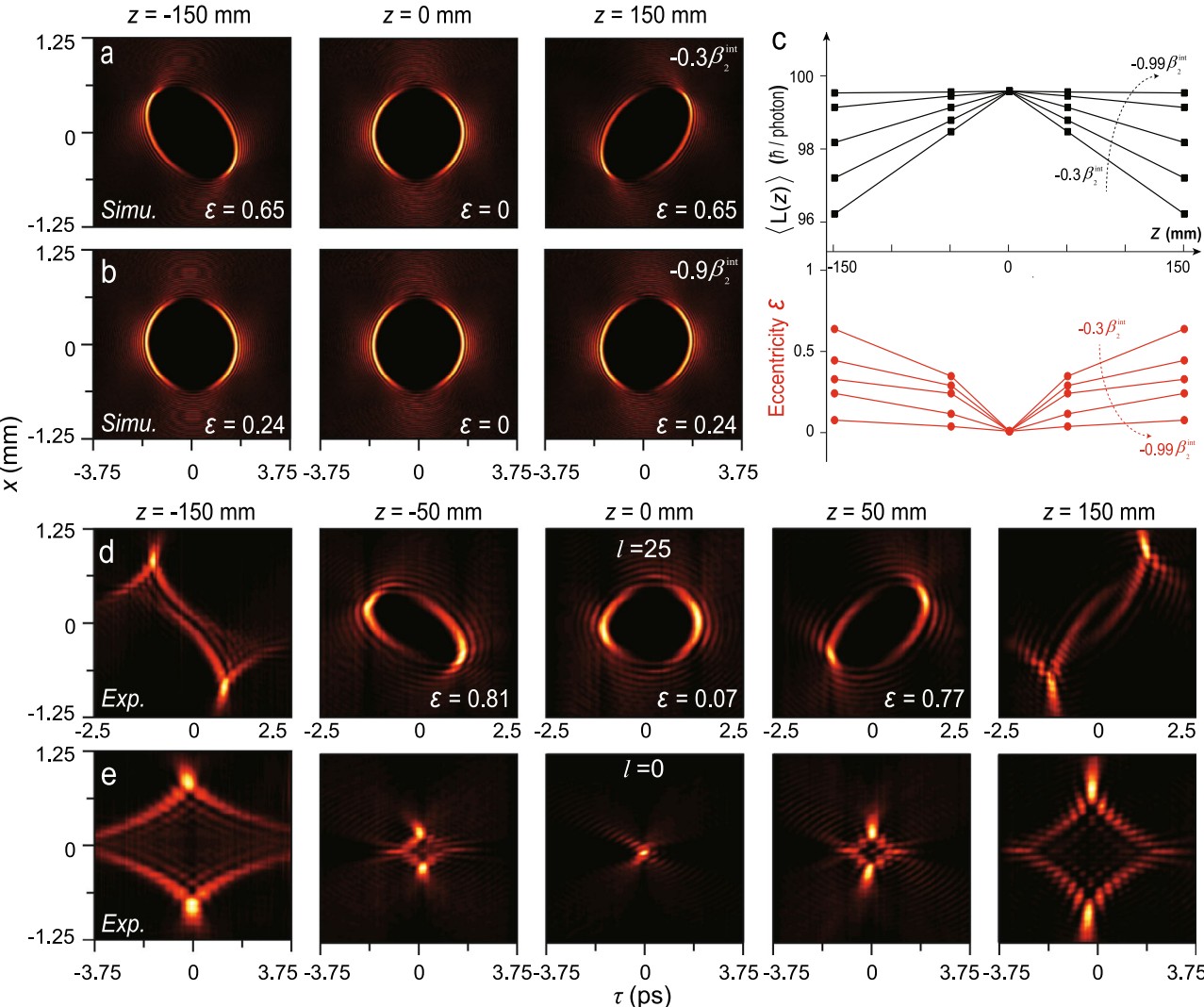

**Fig. 5 Towards time diffraction-free STB vortices. a, b** Simulated evolutions of an STB vortex with $l = 100$ in virtual media with negative dispersion of $-0.3\beta_2^{int}$ and $-0.9\beta_2^{int}$, respectively. Noting that the time diffraction is suppressed to varying degrees compared with Fig. 4d. **c** Calculated integral OAM values and eccentricities depending on the propagating distances in media with negative dispersions of $-0.3\beta_2^{int}$, $-0.5\beta_2^{int}$, $-0.7\beta_2^{int}$, $-0.9\beta_2^{int}$, $-0.99\beta_2^{int}$, where the black and red dashed arrows indicate the direction in which the absolute value of the material dispersion increases. **d, e** Experimental results of propagation dynamics of two STB vortices with topological charges of $l = 25$ and 0, respectively. This indicates that the carried transverse OAM decreases, and the mode evolution accelerates. The corresponding positions are marked at the top of each column in **a**, **b**, **d**, **e**.

observed again, as shown in Fig. 5d. One can clearly see that as the carried transverse OAM decreases, the mode evolution accelerates. At $z = \pm150$ mm, the beam can no longer maintain the singularity of zero intensity. The experimental results of the 0-order STB vortex, i.e., the STB mode, reveal such an accelerating effect more vividly (Fig. 5e). Nevertheless, even for the STB mode, it still maintains ~7 times the Rayleigh distance of a Gaussian beam with the same full-width at half-maximum (Supplementary Note 7). Benefitting from the proposed $x$–$\omega$ modulation, we generate the theoretically equivalent diffraction-free STB beam even though it has already been predicted[29]. The higher stability of higher-order STB vortices also implies that the $k_x$–$\omega$ modulation is the main reason for the difficulty in generating a high-order ST vortex in current experiments, rather than the ST astigmatism effect. Actually, the strong ST coupling quantified by the considerable intrinsic dispersion $\beta_2^{int}$ ensures the identical evolution of STB vortices in most dispersive media.

## Discussion

We have shown that through the immediate $x$–$\omega$ modulation with ingenious phase inverse design, light that carries transverse OAM exceeding $l = 1$ can be controllably generated. In fact, such a way of thinking may also advance research on other time-varying four-dimensional ST beams[27], and inspire applications in conventional beam manipulation[36] and optical encryption[37,38]. In particular, the proposed GDD model is not only applicable to the STB vortices but also to other ST beams wherein the intrinsic dispersion factors can be calculated from their ST spectra accordingly. Given that the scheme of using integrated optics[39,40] to generate ST vortices is still limited by the processing accuracy, material intrinsic absorption, etc., our method may be the first choice for studying transverse OAM at present. At this stage, the SLM in the experiment can be replaced by the flexible metasurfaces[41–43] or the LC geometric phase devices[44,45] with additional polarization response, which would allow the multi-dimensional multiplexing of STB vortices via photonic spin-orbit

interactions. Although our results show that dispersion engineering works well in suppressing the time diffraction, the required negative dispersion is quite large. We expect that similar suppression could be achieved by high-order nonlinear effects (such as self-phase modulation), akin to the formation of optical solitons.

Notably, a cluster of STB vortices with concentric ring ST spectra (i.e., the same $\omega_0$) actually shares the same intrinsic dispersion parameter $\beta_2^{int}$, which means that a Gaussian-like ST vortex—could be seen as the superposition of this series of STB vortices—are subjected to the same time diffraction (Supplementary Discussion). This provides convenience for obtaining time diffraction-free transverse OAM beams using dispersion engineering, nonlinear effects, etc. The research on ST vortices not only expands the possibilities for light field manipulation, i.e., from either spatial or temporal modulation to ST joint modulation, but also enriches the intrinsic physical meaning of OAM, from conventional longitudinal to transverse (even arbitrary) orientation. Due to the inherent ST coupling, such beams bring many nontrivial properties compared with the conventional beam/pulse, such as intrinsic dispersion, time-varying OAM, and transverse spin-orbit interaction. Similar to the longitudinal OAM beam, STB vortices are of great potential to be used in particle manipulation[46], telecommunications[47], and high-dimensional quantum entanglement[48]. Owing to their enhanced robustness enabled by a suppressed time diffraction, novel effects, such as time-varying 'transverse' spin-orbit angular momentum coupling[49], new types of quantum spin Hall effect of light[50], and self-torque of transverse OAM beams in high harmonic generation[51], might be discovered by further studying the interaction of STB vortices and matter. Moreover, due to its simultaneous spatial and temporal (focusing) constraints, the ST vortex has potential for applications in material processing[52].

In conclusion, we have theoretically and experimentally demonstrated the generation of strictly equivalent STB vortices with degradation-free transverse OAM even beyond $10^2$. To the best of our knowledge, this is two orders of magnitude higher than the existing results. Theoretically, this method also can be used to generate higher transverse OAM beams (Supplementary Movies 1–4). This work not only provides direct evidence for photons with ultrahigh transverse OAM but also proposes a universal strategy to generate such OAM. More importantly, we have observed pure time diffraction on transverse OAM beams and accurately described the resulting time-symmetrical evolution by a GDD model with a quantified dispersion factor $\beta_2^{int}$. Based on this, time diffraction-free STB vortices are theoretically proposed with the aid of media dispersion engineering. Our results allow studies and applications of strikingly different physical systems, from light waves, through acoustic waves or other classical waves, to matter waves.

## Methods

**Reconstruction of time-resolved intensity profiles**. A Mach-Zehnder interferometer is utilized to obtain the time-resolved intensities of STB vortices as shown in Fig. 2e, where the reference pulse is obtained from the initial laser pulse via BS$_1$ and temporally reshaped to ~100 fs by a Gaussian spectral filter ($\Delta\lambda = 10$ nm). The STB vortex and the reference beam travel through different optical paths and recombine at BS$_3$. The reference path length is adjusted using a motorized translation stage with steps of 4 μm, 10 μm, 20 μm, and 35 μm for $l = 10, 25, 50,$ and 100. When an STB vortex and a reference pulse overlap in time, the interference fringes are formed and recorded by a CCD camera with a resolution of $1600 \times 1200$ (BGS-USB-SP620, Spiricon). Since the STB vortex is much longer than the reference pulse, whose temporal slices $|E(x, y, \tau)|^2$ can be rebuilt by the interference fringes at a specific time delay $\tau$. Hence, a full three-dimensional STB vortex $|E(x, y, t)|^2$ could be reconstructed by collecting all its temporal slices, as the reference pulse is scanned in the time domain. The reconstructed ST intensities $|E(x, t)|^2$ shown in Fig. 3a–d are obtained by projecting $|E(x, y, t)|^2$ onto the $x$–$t$ plane.

**Reconstruction of phases of STB vortices**. We use the measured interference patterns to reconstruct the phase of an STB vortex. By applying one-dimensional Fourier transform along the $y$ axis to the interference pattern at each time delay $\tau$, we further extract the phase at the peak in the Fourier domain to obtain the one-dimensional phase profile $\phi(x)$ at $\tau$. By stacking the phase patterns at different time delays, the complete phase of an STB vortex can be obtained. Notably, frame-to-frame noise between the phase profiles at different time delays is inevitably introduced by the instability of the interferometer caused by vibrations such as air disturbance. Therefore, to obtain the correct phase, we shifted each phase profile by a specific phase by ensuring the phase at a specific $x_0$ ($x_0$ is away from the STB vortex in each phase profile) is always a constant, i.e., $\phi(x_0, \tau) \equiv$ const. We find that this strategy works well for an STB vortex with a topological charge of less than 25, but for higher order STB vortices, this method is still difficult to overcome the limitations of the instability of the interferometer.

**Numerical simulation**. The simulations are based on a modified forward Maxwell equation, which could be written in the spectral domain[53]:

$$\frac{\partial \tilde{E}}{\partial \zeta} = \frac{i}{2k(\omega)}\Delta_\perp \tilde{E} + i[k(\omega) - \nu(\omega)]\tilde{E}, \qquad (6)$$

where $\Delta_\perp \equiv \partial^2/\partial x^2$ is the transverse Laplacian operator, $\tilde{E} = \tilde{E}(x, \Omega)$ is the pulse complex envelope, $k(\omega)$ is the whole wavevector, $\nu(\omega) = k_0 + (\omega - \omega_0)/v_g$ is the reduction wavevector where $v_g$ is the group velocity, and $\zeta$ relates to the pulse frame $\partial_\zeta = \partial_z - \omega/v_g$. We solved Eq. (6) numerically with the parameters the same as their experimental values. In the air propagation case, the whole wavevector could be calculated by $k(\omega) \approx n_{air}(\omega)\omega/c$, where $n_{air}(\omega)$ is the refractive index of air. In the dispersion media case, we used $k(\omega) \approx k_0 + 1/v_g(\omega - \omega_0) + (1/2)\beta_2(\omega - \omega_0)^2$ to construct the equivalent dispersion medium.

**Calculation for integral OAM values**. The integral OAM values of STB vortices are calculated through a volume integral as described in ref. [1,14]:

$$\langle \mathbf{L} \rangle = 1/\varepsilon_0 \omega_0 \int \mathbf{r} \times \mathbf{p} \, dV / \left( \int E^* E \, dV \right), \qquad (7)$$

where the unit of $\langle \mathbf{L} \rangle$ is $\hbar$ per photon, $\tilde{E} = \tilde{E}(x, \Omega)$ is the pulse complex envelope, $\varepsilon_0$ is the dielectric constant, $\mathbf{r} \times \mathbf{p}$ donates the transverse OAM density where $\mathbf{r}$ is the position vector and $\mathbf{p}$ is the linear momentum density.

## Data availability

The data that support the plots within this paper and other findings of this study are available from the corresponding authors upon reasonable request.

## Code availability

The codes that support the findings of this study are available upon reasonable request from the corresponding authors.

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

## Acknowledgements

The work of W.C., W.Z., Y.L., and Y.Q.L was supported by the National Key Research and Development Program of China (2017YFA0303700) and the Natural Science Foundation of Jiangsu Province, Major Project (BK20212004). The work of F.C.M and J.M.D was supported by the Contract ANR-15-IDEX-0003 and ANR-17-EURE-0002.

## Author contributions

W.C. and W.Z. proposed the original idea. W.C. performed all experiments and some theoretical analysis. W.Z. performed all theoretical analysis and some experiments. Y.L. contributed to developing the measurement method. F.C.M contributed to the simulation. Y.Q.L and J.M.D guided the theoretical analysis and supervised the project. All authors contributed to writing the manuscript.

## Competing interests

The authors declare no competing interests.
