## [Peer Review File · Nature Communications]

Time diffraction-free transverse orbital angular momentum beamsREVIEWER COMMENTS

Reviewer #1 (Remarks to the Author):

Manuscript title: Time diffraction-free transverse orbital angular momentum beams

Authors: Wei Chen, Wang Zhang, Yuan Liu, Fan-Chao Meng, John M. Dudley, Yan-Qing

In this work the authors analyze both theoretically and experimentally the spatiotemporal coupling of beams carrying transverse orbital angular momentum. It is shown that the mode degradation can be avoided or vastly minimized by a suitable modulation in the $x-\omega$ plane. The procedure for the inverse design of the required phase is described and is well illustrated. In this way the authors generate equivalent spatiotemporal Bessel vortices carrying transverse OAM of the order of 100. This is an interesting manuscript containing important new information. In this sense, the manuscript deserves publication.

On the other hand, this manuscript is not easy to read. I think this is primarily due to its structure. In my opinion, parts essential to understanding it are left in the Supplement. From the point of view of the experimenters, the manuscript, in various places, lacks essential information. The paper is not "engineering-friendly" as claimed on page 6 of the manuscript. As such, I recommend a major technical revision of the manuscript in the following sense.

/1/ In page 4 the authors briefly mention the experimental setup and redirect the reader to Supplementary Fig. S3. In my opinion, the place of the experimental setup is here and not as Fig. S3. It should be described to be informative for the broad audience by saying what is the number of pixels of the SLM, the dimension of the individual pixel, the focal length of the cylindrical lens, how many lines per unit length are on the diffraction grating, the distances between the elements, what are the essential parameters of the CD camera, the size of the diaphragm A (which is probably a 1-D slit). From the description of the inverse design of spiral phase it appear that binarized phase distributions are projected on these SLM. If so, this has to be clearly stated as well.

/2/ In the same paragraph the authors state that "Besides, the phase reconstruction of high-order modes is difficult due to air disturbance-induced noise and the limited accuracy of the equipment in the experiment." This means nothing because practically nothing is said about the experiment.

/3/ In page 5, line 6 after Eq. 3, the statement "... since the pre-chirp is completely compensated after a short propagation of ~ 150 mm, the corresponding dispersion factor" is not informative. Nothing is said about the duration of the input femtosecond pulses.

/4/ In page 5 the statement "...unlike the longitudinal OAM beam, for which the sign of the topological charge is difficult to judge just by the intensity distribution..." has to be softened. Please remember the situation when focusing such a beam with a simple cylindrical lens. According to my experience, the method works fine for topological charges of the order of 30.

/5/ Regarding Fig. S1: In the left panel, the x-axis is horizontal. In the four frames on the right, it is the vertical axis. Perception of the figure would be somewhat easier if the left panel were rotated at 90 degrees (the captions, too).

/6/ Regarding Fig. S3: My recommendation to move the figure to the main text remains. Also, it makes sense to add the SLM sketched so that the spatial x-axis and spectral c-axis can be denoted. In the explanation the authors may wish to clearly say that at the output of the MZI a series of interference pictures at an adjustable introduced delay in its arm are recorded and from these pictures they obtain what the authors call "... simulated intensities (spiral phases)". Citing only [4,5] does not make the manuscript self-contained.

/7/ Regarding Fig. S4: Does "measured intensities" mean "measured interference patterns"? If so, please state this clearly. If no, what is the method used for the phase reconstruction. The results are interesting, but in order to be convincing, the reader has to understand how they are obtained.

General comment: The use of the adjective "instantaneous" in the manuscript disturbs me. In fact, a stationary-in-time two-dimensional phase modulation is implemented with a two-dimensional spatial phase modulator, which is not a fast device. Perhaps the authors would consider another term or comment in more detail on what they mean by "instantaneous". In the four Supplementary videos it would be particularly useful to see also x-y-projections for the different local times. This is just a personal opinion, as it would be very useful, but would probably be too time-consuming for the authors.

Possible typographical problem on p. 2, line 10 from top: "contributed" or "attributed"?

Reviewer #2 (Remarks to the Author):

This manuscript reports on the experimental demonstration and theoretical study of Bessel-type spatiotemporal (ST) vortices with topological charges up to 100 and beyond. The ST vortices carrying transverse orbital angular momentum (OAM) are recently focused and placed great expectations to considerably extend the applications of conventional OAM beams. However, such potential is largely limited by previous difficulties in generating high transverse OAM due to mode degradation. Existing experiments were limited in low transverse OAM, e.g., $l = 1, 2$. The authors verify the main reason is the time-delayed modulation relying on a spatial Fourier transform, and give an effective solution by introducing an instantaneous $x-\omega$ modulation via the inverse design of phase. This paves the way to achieve ST vortices with large OAM, which is of great value for promoting the real-world applications. I think this work is of considerable interest to the community and worth publishing in Nature Communications, but I recommend the authors consider the following points to improve the manuscript:

1. The authors should clarify why the value of γ is specially chosen to satisfy the requirement that the Bessel ST vortex is a circularly symmetric pulse.
2. Does the polarization of the initial light need to be adjusted during beam synthesis, and will it affect the results?
3. The methods for pulse phase reconstruction should be mentioned. In addition, although the 3D profiles of vortices are impressive, the recovery scheme is not well presented. The authors could provide the details in the supplementary information.
4. The authors point out the spatial or temporal diameter dependence on the topological charges of ST vortices. As a supplement, the relationship between the diameter and spectral bandwidth also should be clarified.
5. The proposed GDD model is quite effective. I note that this model seems to be applicable to other types of ST beams as well. It is recommended to comment on this point.
6. The authors comment on the synthesis of ST vortices via integrated optical schemes, and they may want to comment on the feasibility of generation using metasurfaces which have been widely used to manipulate various types of optical angular momentum, such as Jin et al, eLight 1, 5 (2021), Ni et al Science 374, eabj0039 (2021), Ouyang et al, Nature Photonics 15, 901 (2021), etc
7. The dispersion parameter β shown in Fig. 4b does not agree with the value given in the corresponding explanatory text.

Reviewer #3 (Remarks to the Author):

The manuscript describes a theoretical and experimental study of transverse OAM beams. This is an interesting subject which helps our understanding of solutions of Maxwell's equations. Here, a specific superposition of plane waves is considered where the plane waves are chosen such that a ST vortex is present in the superposition. The manuscript introduces well the background of the subject and concentrates on explaining an experimental procedure to create these beams in a stable way. The subject is interesting and suitable for publication.

To reach a wider audience, it would be helpful to highlight the physical significance of the ST vortex and how its presence changes the properties of the beam/pulse. In this context, the language needs to be more precise. For example, the abstract speaks about "space-time coupling", however the equations considered are linear and there is no coupling between the different plane waves considered. All we observe is an interference pattern. The main question then translates to identifying a field property such as energy, momentum, angular momentum, or chirality which is specific to the ST vortex beams. For example, plane waves are defined by a given energy and linear momentum and standard OAM beams are defined by a given energy, direction, and angular momentum (spin and OAM). The ST vortex beams/pulses need to be defined through other physical meaningful properties.

Response to reviewer comments

Reviewer 1

In this work the authors analyze both theoretically and experimentally the spatiotemporal coupling of beams carrying transverse orbital angular momentum. It is shown that the mode degradation can be avoided or vastly minimized by a suitable modulation in the $x-\omega$ plane. The procedure for the inverse design of the required phase is described and is well illustrated. In this way the authors generate equivalent spatiotemporal Bessel vortices carrying transverse OAM of the order of 100. This is an interesting manuscript containing important new information. In this sense, the manuscript deserves publication.

On the other hand, this manuscript is not easy to read. I think this is primarily due to its structure. In my opinion, parts essential to understanding it are left in the Supplement. From the point of view of the experimenters, the manuscript, in various places, lacks essential information. The paper is not "engineering-friendly" as claimed on page 6 of the manuscript. As such, I recommend a major technical revision of the manuscript in the following sense.

Author Response:

We thank the reviewers for the positive appraisal and valuable suggestions of our work. In the following, we have made technical modifications to the updated manuscript, which we believe improve the readability of the paper.

1. In page 4 the authors briefly mention the experimental setup and redirect the reader to Supplementary Fig. S3. In my opinion, the place of the experimental setup is here and not as Fig. S3. It should be described to be informative for the broad audience by saying what is the number of pixels of the SLM, the dimension of the individual pixel, the focal length of the cylindrical lens, how many lines per unit length are on the diffraction grating, the distances between the elements, what are the essential parameters of the CD camera, the size of the diaphragm A (which is probably a 1-D slit). From the description of the inverse design of spiral phase it appear that binarized phase distributions are projected on these SLM. If so, this has to be clearly stated as well.

Author Response:

In the revised manuscript, we have combined the phase patterns for generating STB vortices (in the original manuscript) and the experimental devices (in original Supplementary Fig. S3), and have moved them to the main text (Fig. 2 in the revised manuscript), as shown in Fig.R1. The coordinate in the upper left corner of Fig. R1e marks the orientations in SLM where the phase patterns are loaded, which helps to describe how the phase patterns are loaded. We have also added the details of the experimental setup, especially the specific parameters of the SLM, the pulse duration of the incident pulse, the focal length of cylindrical lens L_{y1} which determines the distances between the elements, and the size of the diaphragm A (remarked as the "1D slit" in the updated manuscript). Additionally, the details of the reconstruction for the space-time intensity distribution of the STB vortices (including the parameters of the CCD camera) have been added to the "Methods" section in the main text for the convenience of the readers.

Figure R1| SLM phase patterns and experimental setup for generating and characterizing STB vortices. a–d, Phase patterns of STB vortices with topological charges of $l = 10, 25, 50,$ and 100 . e, Experimental setup which consists of two sections: (1) STB vortex generator consisting of a grating, a cylindrical lens L_{y1} , an aperture, and an SLM; and (2) a time-resolved profile analyser that is realized by a Mach–Zehnder interferometer consisting of two BSe (BS₁ and BS₃), a CCD camera, and a motorized translation stage in the reference path. The coordinate in the upper left corner of e marks the orientations in SLM where the phase patterns in a–d are loaded.

We have added or revised the following sentences to the main text:

“To generate STB vortices, we used a custom 4f-pulse shaper consisting of a diffraction grating (1800 lines/mm, GH25-18V, Thorlabs), a cylindrical lens (L_{y1} with a focal length of $f = 100$ mm, which also determines the distances between the elements), and an LC-based 2D phase-only SLM (PLUTO-2.1-NIR-133, Holoeye) with a resolution of 1920×1080 , a pixel pitch of $\sim 8 \mu\text{m}$, and an active area of $\sim 15.36 \times 8.64$ mm (Fig. 2e). The frequencies of an ultrashort pulse that comes from a Ti:sapphire laser with a central wavelength of ~ 800 nm and a pulse duration of ~ 35 fs are spatially spread by the grating and collimated to the SLM via the L_{y1} , which can be understood as a temporal Fourier transform. We consider the SLM as the x - ω plane, where the phase patterns for the generation of STB vortices are loaded. After the SLM, the light field is retroreflected and reconstituted at the same diffraction grating, thereby immediately generating the STB vortices, with no need for a time-delayed SFT. The ST intensities were measured and reconstructed using a Mach-Zehnder interferometer shown in Fig. 2e (Methods)^{33,34}.” (page 5, paragraph 1)

“Notably, the spatial and temporal bandwidths were set to $\Delta k_x = \sim 123$ rad/mm (by the phase patterns) and $\Delta \lambda = \sim 12$ nm (by the one-dimensional slit with a width of ~ 5 mm (Fig. 2e)) for temporal and spatial scale consistency, ...” (page 5, paragraph 2)

“Reconstruction of time-resolved intensity profiles. A Mach-Zehnder interferometer is utilized to obtain the time-resolved intensities of STB vortices as shown in Fig. 2e, where the reference

pulse is obtained from the initial laser pulse via BS_1 and temporally reshaped to ~ 100 fs by a Gaussian spectral filter ($\Delta\lambda = 10$ nm). The STB vortex and the reference beam travel through different optical paths and recombine at BS_3 . The reference path length is adjusted using a motorized translation stage e with steps of $4\ \mu\text{m}$, $10\ \mu\text{m}$, $20\ \mu\text{m}$, and $35\ \mu\text{m}$ for $l = 10, 25, 50,$ and 100 . When an STB vortex and a reference pulse overlap in time, the interference fringes are formed and recorded by a CCD camera with a resolution of 1600×1200 (BGS-USB-SP620, Spiricon). Since the STB vortex is much longer than the reference pulse, whose temporal slices $|E(x, y, \tau)|^2$ can be rebuilt by the interference fringes at a specific time delay τ . Hence, a full three-dimensional STB vortex $|E(x, y, t)|^2$ could be reconstructed by collecting all its temporal slices, as the reference pulse is scanned in the time domain. The reconstructed ST intensities $|E(x, t)|^2$ shown in Figs. 3a–d are obtained by projecting $|E(x, t)|^2$ onto the x – t plane.” (the “**Methods**” section)

2. In the same paragraph the authors state that "Besides, the phase reconstruction of high-order modes is difficult due to air disturbance-induced noise and the limited accuracy of the equipment in the experiment." This means nothing because practically nothing is said about the experiment.

Author Response:

We apologize for not providing enough valid information for the phase reconstruction. We use the measured interference patterns (from the Mach-Zehnder interferometer) to reconstruct the phase of an STB vortex. By applying one-dimensional Fourier transform along the y axis to the interference pattern at each time delay τ , we further extract the phase at the peak in the Fourier domain to obtain the one-dimensional phase profile $\phi(x)$ at τ . By stacking the phase patterns of different time delays, the complete phase of an STB vortex can be obtained. Notably, frame-to-frame noise between the phase profiles at different time delays is inevitably introduced by the instability of the interferometer caused by vibrations such as air disturbance. Therefore, to obtain the correct phase, we shifted each phase profile by a specific phase by ensuring the phase at a specific x_0 (x_0 is away from the STB vortex in each phase profile) is always a constant, *i.e.*, $\phi(x_0, \tau) \equiv \text{const}$. We find that this strategy works well for an STB vortex with a topological charge of less than 25, but for higher order STB vortices, this method is still difficult to overcome the limitations of the instability of the interferometer. Nevertheless, the simulated phases (Figs. 3e–h) verify the spiral phases of the corresponding topological charges and the carried transverse OAM (see also the reconstructed phases for $l = 3, 5, 10,$ and 25 in Supplementary Fig. S3).

We now include the phase reconstruction strategy in the “**Methods**” section and change “air disturbance-induced noise and the limited accuracy of the equipment in the experiment” in the original manuscript to “the instability of the interferometer caused by vibrations such as air disturbance” (see **page 5, paragraph 3**), for a clearer expression.

3. In page 5, line 6 after Eq. 3, the statement "... since the pre-chirp is completely compensated after a short propagation of ~ 150 mm, the corresponding dispersion factor" is not informative. Nothing is said about the duration of the input femtosecond pulses.

Author Response:

We have now added the duration of the input femtosecond pulses (~ 35 fs) in the description of the experiment setup in the revised manuscript (see **page 5, paragraph 1**). We also note that the statement here may have caused a misunderstanding among the reviewers, and we would like to emphasize that the STB vortex with the pre-chirp (~ -0.231 ps²) converts to a standard non-chirped STB vortex after only propagating ~ 150 mm in the simulation (Fig. 4d), which means that this pre-chirp has been completely compensated in such a short distance of propagation. Hence, here we are more concerned with the shape of the STB vortex on the $x-t$ plane rather than (just) the pulse duration.

We have also changed the original statement "... we observed consistent results in the simulation (Fig. 3c) ...since the pre-chirp is completely compensated after a short propagation of ~ 150 mm, the corresponding dispersion factor..." to "we observed consistent results in the simulation (Fig. 4d), which means the pre-chirp is completely compensated after a short distance of propagation from $z = -150$ mm (the elliptical chirped STB vortex) to $z = 0$ mm (the standard non-chirped STB vortex). At first, we attributed this to the positive dispersion of the air, however, the corresponding dispersion factor is ~ 1.542 ps²/m...", to make the statement more precise. (see **page 6, paragraph 3**)

4. In page 5 the statement "...unlike the longitudinal OAM beam, for which the sign of the topological charge is difficult to judge just by the intensity distribution..." has to be softened. Please remember the situation when focusing such a beam with a simple cylindrical lens. According to my experience, the method works fine for topological charges of the order of 30.

Author Response:

We have changed this statement to "unlike the longitudinal OAM beam, for which the sign of the topological charge is difficult to judge without extra devices, such as the cylindrical lens, the results with $l = -100$ are distinguishable because they look like the mirror version of $l = 100$ (Fig. 4b)" in **page 6, paragraph 2**, which makes the statement more precise.

5. Regarding Fig. S1: In the left panel, the x -axis is horizontal. In the four frames on the right, it is the vertical axis. Perception of the figure would be somewhat easier if the left panel were rotated at 90 degrees (the captions, too).

Author Response:

Now the left panel in Fig. S1 has been rotated at 90 degrees, see also Fig. R2.

Figure R2 | Principle of inverse design of phase. **a**, Principle. **b–e**, Phase patterns for generating STB vortices with different topological charges of $l = 0, 10, 11$ and 20 . The white dashed lines in **b–e** mark the positions of the left and right main lobes, and the blue arrows point out the accumulated dislocation of phase between the left and right main lobes.

6. Regarding Fig. S3: My recommendation to move the figure to the main text remains. Also, it makes sense to add the SLM sketched so that the spatial x -axis and spectral c -axis can be denoted. In the explanation the authors may wish to clearly say that at the output of the MZI a series of interference pictures at an adjustable introduced delay in its arm are recorded and from these pictures they obtain what the authors call "... simulated intensities (spiral phases)". Citing only [4,5] does not make the manuscript self-contained.

Author Response:

The experimental setup in original Fig. S3 has now been moved to the main text (see **Fig. 2e**). The coordinate in the upper left corner of Fig. 2e marks the orientations in SLM where the phase patterns are loaded, which helps to describe how the phase patterns are loaded. Besides, the detailed scheme of using MZI to reconstruct the ST intensity distribution and phase of STB vortices has been added to the “**Methods**” section.

7. Regarding Fig. S4: Does "measured intensities" mean "measured interference patterns"? If so, please state this clearly. If no, what is the method used for the phase reconstruction. The results are interesting, but in order to be convincing, the reader has to understand how they are obtained.

Author Response:

We apologize for the ambiguous expression. The “measured intensities” does not mean “measured interference patterns”, but the “reconstructed intensities” of STB vortices. We have replaced these ambiguous descriptions with “reconstructed intensities” in the revised manuscript. As we responded above, the detailed scheme for the reconstruction of the ST intensity distribution and the phase of STB vortices has been added to the “**Methods**” section.

General comment: The use of the adjective "instantaneous" in the manuscript disturbs me. In fact, a stationary-in-time two-dimensional phase modulation is implemented with a two-dimensional spatial phase modulator, which is not a fast device. Perhaps the authors would consider another term or comment in more detail on what they mean by "instantaneous".

Author Response:

We totally agree with the reviewer's consideration. We use "instantaneous" to emphasize the difference from the time-delayed $k_x\text{-}\omega$ modulation employed in the previous experiments, where "instantaneous" indicates that the spatial Fourier transform is applied immediately on the modulation plane, with no need for extra propagation. Therefore, one can use the conventional liquid-crystal (LC) based SLM to accomplish such modulation, although the response is not fast. We understand the reviewer's concerns, and in the revised manuscript, we have replaced "instantaneous" with "immediate" and provided some explanations, which may resolve the confusion of the reviewer and readers. We have added this description in **page 4, paragraph 1**.

In the four Supplementary videos it would be particularly useful to see also x - y -projections for the different local times. This is just a personal opinion, as it would be very useful, but would probably be too time-consuming for the authors.

Author Response:

We have added the x - y projections of STB vortices for the different local times in the revised **Supplementary videos S1–S4**.

Possible typographical problem on p. 2, line 10 from top: "contributed" or "attributed"?

Author Response:

We have corrected this typo.

Reviewer 2

This manuscript reports on the experimental demonstration and theoretical study of Bessel-type spatiotemporal (ST) vortices with topological charges up to 100 and beyond. The ST vortices carrying transverse orbital angular momentum (OAM) are recently focused and placed great expectations to considerably extend the applications of conventional OAM beams. However, such potential is largely limited by previous difficulties in generating high transverse OAM due to mode degradation. Existing experiments were limited in low transverse OAM, *e.g.*, $l = 1, 2$. The authors verify the main reason is the time-delayed modulation relying on a spatial Fourier transform, and give an effective solution by introducing an instantaneous $x\text{-}\omega$ modulation via the inverse design of phase. This paves the way to achieve ST vortices with large OAM, which is of great value for promoting the real-world applications. I think this work is of considerable interest to the community and worth publishing in Nature Communications, but I recommend the authors consider the following points to improve the manuscript:

Author Response:

We thank the reviewer for their positive appraisal of our work, and for the clear summary of the accomplishments realized in this work.

1. The authors should clarify why the value of γ is specially chosen to satisfy the requirement that the Bessel ST vortex is a circularly symmetric pulse.

Author Response:

According to ref. [R1], to make the integral OAM value $l\hbar$ per photon, the value of γ (~ 3.48 ns/m in our experiment) should satisfy the requirement that the STB vortices are circularly symmetric pulses with equal width and length when the temporal t axis is projected to the spatial z axis, and we call this “temporal and spatial scale consistency”. Otherwise, the OAM value will become larger. We have added this discussion in **page 5, paragraph 2**.

2. Does the polarization of the initial light need to be adjusted during beam synthesis, and will it affect the results?

Author Response:

To ensure that SLM does not affect the polarization, we adjusted the incident light to be linearly polarized in the y direction, which means that the polarization of the generated STB vortices is perpendicular to the $x-t$ plane. According to the theoretical analysis in ref. [R1], this out-of-plane polarization will result in a zero intensity at the center of STB vortices (verified by Figs. 3a–d). In contrast, the in-plane polarization will lead to an observable nonzero intensity in the center of STB vortices due to transverse spin-orbit interaction [R1]. We have added this discussion in **page 5, paragraph 2**.

3. The methods for pulse phase reconstruction should be mentioned. In addition, although the 3D profiles of vortices are impressive, the recovery scheme is not well presented. The authors could provide the details in the supplementary information.

Author Response:

The detailed scheme for the reconstruction of the ST intensity distribution (including the 3D iso-surface profile) and the phase of STB vortices has been added to the “**Methods**” section.

4. The authors point out the spatial or temporal diameter dependence on the topological charges of ST vortices. As a supplement, the relationship between the diameter and spectral bandwidth also should be clarified.

Author Response:

We find that the widths of the STB vortices on the x and t axes are inversely proportional to the spatial and temporal spectral bandwidths, respectively, which are consistent with conventional spatial beams and short pulses. Actually, the measured similar widths of the STB vortices on the $x-t$ plane (with $l = 5$ and 10) in the original Supplementary Fig. S4 have verified this point (see also Fig. R3). In the revised manuscript, we have added this discussion in **page 5, paragraph 3**, and have also emphasized this in **Supplementary Fig. S3**.

Figure R3 | Reconstructed intensities and phases of STB vortices with $l=3, 5, 10,$ and 25 . a–b, Reconstructed intensities. e–h, Reconstructed phases accordingly. The white circles in e–h mark the spiral phases of each STB vortex. The corresponding topological charges are marked at the top of each column.

5. The proposed GDD model is quite effective. I note that this model seems to be applicable to other types of ST beams as well. It is recommended to comment on this point.

Author Response:

Indeed, we note that the proposed GDD model is not only applicable to the special STB vortices but also to other ST beams wherein the intrinsic dispersion factors can be calculated from their ST spectra accordingly. We have added this discussion in **page 8, paragraph 3**.

6. The authors comment on the synthesis of ST vortices via integrated optical schemes, and they may want to comment on the feasibility of generation using metasurfaces which have been widely used to manipulate various types of optical angular momentum, such as Jin et al, *eLight* 1, 5 (2021), Ni et al *Science* 374, eabj0039 (2021), Ouyang et al, *Nature Photonics* 15, 901 (2021), *etc.*

Author Response:

Although it is still very challenging to generate STB vortices directly using integrated optics at this stage, as considered by the reviewer, the SLM used in the current experiment can be replaced with metasurfaces, thereby realizing multi-dimensional manipulation of STB vortices through photonic spin-orbit interactions. We note that the liquid crystal geometric phase devices can also realize similar polarization responses. We have added this discussion in **page 8, paragraph 3**, along with the suggested references.

7. The dispersion parameter β shown in Fig. 4b does not agree with the value given in the corresponding explanatory text.

Author Response:

We have corrected this typo.

Reviewer 3

The manuscript describes a theoretical and experimental study of transverse OAM beams. This is an interesting subject which helps our understanding of solutions of Maxwell's equations. Here, a specific superposition of plane waves is considered where the plane waves are chosen such that a ST vortex is present in the superposition. The manuscript introduces well the background of the subject and concentrates on explaining an experimental procedure to create these beams in a stable way. The subject is interesting and suitable for publication.

Author Response:

We thank the reviewer for their supportive comments.

To reach a wider audience, it would be helpful to highlight the physical significance of the ST vortex and how its presence changes the properties of the beam/pulse.

Author Response:

We thank the reviewer for their valuable suggestion. An ST vortex can be understood physically as a propagating vortex structure where the vortex opening is perpendicular to the direction of propagation. Such transverse vortices are, in fact, common in nature and science and can be seen in diverse systems such as the movement of boomerangs used by the First Nation Australian peoples, and in the dynamics of the tropical cyclone, the human heart, and magnetic nanowires [R2-R5]. In optics, research on ST vortices not only expands the possibilities for light field manipulation, *i.e.*, from either spatial or temporal modulation to ST joint modulation, but also enriches the intrinsic physical meaning of OAM, from conventional longitudinal to transverse (even arbitrary) orientation. Due to the inherent ST coupling, the properties of the propagating field change significantly: such beams possess many nontrivial properties when compared with the conventional beam/pulse, such as intrinsic dispersion, time-varying OAM, and transverse spin-orbit interaction. Moreover, due to its simultaneous spatial and temporal (focusing) constraints, the ST vortex has potential for applications in material processing [R6]. Therefore, the ST vortex is physically significant and worthy of further exploration. We have added this discussion in **page 2, paragraph 1 and page 9, paragraph 1** and have added additional references.

In this context, the language needs to be more precise. For example, the abstract speaks about “space-time coupling”, however the equations considered are linear and there is no coupling between the different plane waves considered. All we observe is an interference pattern.

Author Response:

We have rechecked and revised some descriptions in the updated manuscript to make the language more precise. Indeed, an STB vortex can be considered as the interference pattern of different plane waves, and there is no “explicit” coupling between different plane waves. Nevertheless, the “space-time coupling” here refers to the one-to-one correspondence between spatial frequency k_x and temporal frequency ω (for STB vortices, one sees such coupling clearly from Eq. (1) in the manuscript, *i.e.*, $\tilde{E}(k_x, \Omega) = \tilde{E}_l(\kappa, \varphi) = \delta(\kappa - R_0)e^{i\ell\varphi}$). Based on this special ST frequency-frequency relationship, we derived the intrinsic dispersion factor β_2^{int} of the STB vortex, which helps us understand its unique propagation dynamics. In fact, our usage

of this terminology of “space-time coupling” is in line with that which is used in the literature to describe similar ST spectral relationships within other ST beams [R7-R9]. For the ST beam with such modulated ST spectra, changing the spectral properties of the pulse also changes its spatial properties and vice versa. To make it clearer, we have added this discussion in **page 3, paragraph 2** and **page 4, paragraph 1** and have added additional references.

The main question then translates to identifying a field property such as energy, momentum, angular momentum, or chirality which is specific to the ST vortex beams. For example, plane waves are defined by a given energy and linear momentum and standard OAM beams are defined by a given energy, direction, and angular momentum (spin and OAM). The ST vortex beams/pulses need to be defined through other physical meaningful properties.

Author Response:

We totally agree with the reviewer regarding their heuristic thinking about the ST vortex. To date, studies on ST vortices have mainly focused on their ST distributions and the similarities to spatial vortices while ignoring their physical differences, to a certain extent. In this work, we partially revealed the physical particularity of the ST vortex, such as the time-symmetric evolution induced by intrinsic dispersion β_2^{int} . It is worth mentioning that the transverse OAM carried by the ST vortex is time-varying during propagation (as shown in Fig. 5c). In the revised manuscript, we further provide the approximate fitted expression of such time-varying OAM for $l=100$ from the numerical results, *i.e.*, $\langle L(z) \rangle \approx \{1 - [\omega_0^2 \alpha(l)/4\pi^2](\beta_2^{\text{int}} + \beta_2^{\text{ext}})|z|\}l$, where l is the OAM value of standard STB vortices, $\alpha(l)$ is a dimensionless constant related to l where $\alpha(l = 100) \approx \sqrt{2} \times 10^{-6}$, β_2^{ext} is the media dispersion, and $|z| = |ct|$ is the propagation distance. We consider that $\langle L(z) \rangle$ may be a special physical quantity to define ST vortices because it indicates that their time-varying OAM fundamentally depends on the intrinsic dispersion factor β_2^{int} . Additionally, it is possible to obtain a more accurate analytical expression of $\langle L(z) \rangle$ from Maxwell's equations, but this is beyond the scope of this work. Our results may inspire us and others to work in this crucial direction, thereby finding other unique physical quantities of the ST vortices in the future. We have added this discussion in **page 7, paragraph 3**.

References:

- [R1] K. Y. Bliokh, Spatiotemporal vortex pulses: angular momenta and spin-orbit interaction. *Phys. Rev. Lett.* **126**, 243601 (2021).
- [R2] P. Peduzzi, et al. Global trends in tropical cyclone risk. *Nature Clim. Change* **2**, 289–294 (2012).
- [R3] G. R. Hong, et al. Characterization and quantification of vortex flow in the human left ventricle by contrast echocardiography using vector particle image velocimetry. *JACC Cardiovasc. Imaging* **1**, 705–717 (2008).
- [R4] S. S. P. Parkin, M. Hayashi, & L. Thomas, Magnetic domain-wall racetrack memory. *Science* **320**, 190–194 (2008).
- [R5] I. M. Andersen, et al. Exotic transverse-vortex magnetic configurations in CoNi nanowires.

ACS Nano **14**, 1399–1405 (2020).

- [R6] W. Cheng, X.-L. Liu, & P. Polynkin, Simultaneously spatially and temporally focused femtosecond vortex beams for laser micromachining. *J. Opt. Soc. Am. B* **35**, B16–B19 (2018).
- [R7] H. E. Kondakci, & A. F. Abouraddy, Diffraction-free space–time light sheets. *Nat. Photon.* **11**, 733–740 (2017).
- [R8] H. E. Kondakci, & A. F. Abouraddy, Optical space-time wave-packets having arbitrary group velocities in free space. *Nat. Commun.* **10**, 929 (2019).
- [R9] A. Forbes, M. de Oliveira, & M. R. Dennis, Structured light. *Nat. Photon.* **15**, 253–262 (2021).

Other modifications

We have reformatted the equations with *Latex* in the revised manuscript and supplementary information.

REVIEWERS' COMMENTS

Reviewer #1 (Remarks to the Author):

I had the pleasure of reading a substantially revised manuscript. In it, my critical comments are adequately addressed, including the one about the supplementary videos. In my opinion, the manuscript has been substantially improved and I recommend to accept it for publication in Nature Communications.

Reviewer #2 (Remarks to the Author):

I read through the response and revision. The authors not only addressed my concerns well but also answered the comments from other referees well. Hence, i don't have any reasons to hold this work from being accepted. a high-quality and innovative job!

Reviewer #3 (Remarks to the Author):

I thank the authors for following up on the suggestions and highlighting their definition of coupling as being a relationship between quantities. While the time dependent OAM is not a conserving characteristic it is a good idea to include this property and its approximate expression. I agree with the authors that this hopefully helps the further study of these properties. I therefore consider this manuscript to be publishable and of interest to the community.

Response to reviewer comments

Reviewer 1

I had the pleasure of reading a substantially revised manuscript. In it, my critical comments are adequately addressed, including the one about the supplementary videos. In my opinion, the manuscript has been substantially improved and I recommend to accept it for publication in Nature Communications.

Author Response:

We thank the reviewer for their positive comments and recommendation for publication.

Reviewer 2

I read through the response and revision. The authors not only addressed my concerns well but also answered the comments from other referees well. Hence, i don't have any reasons to hold this work from being accepted. a high-quality and innovative job!

Author Response:

We thank the reviewer for their positive comments and recommendation for publication.

Reviewer 3

I thank the authors for following up on the suggestions and highlighting their definition of coupling as being a relationship between quantities. While the time dependent OAM is not a conserving characteristic it is a good idea to include this property and its approximate expression. I agree with the authors that this hopefully helps the further study of these properties. I therefore consider this manuscript to be publishable and of interest to the community.

Author Response:

We thank the reviewer for their positive comments and recommendation for publication.